# The impact of end-demand legislation on sex workers' access to health and sex worker-led services: A community-based prospective cohort study in Canada

Elena Argento[1,2]☯, Shira Goldenberg[1,3]‡, Melissa Braschel[1]‡, Sylvia Machat[1]‡, Steffanie A. Strathdee[4]‡, Kate Shannon[1,2]☯*

1 Centre for Gender & Sexual Health Equity, Vancouver, British Columbia, Canada, 2 Faculty of Medicine, University of British Columbia, Vancouver, British Columbia, Canada, 3 Faculty of Health Sciences, Simon Fraser University, Burnaby, British Columbia, Canada, 4 Department of Medicine, University of California San Diego, La Jolla, California, United States of America

☯ These authors contributed equally to this work.
‡ These authors also contributed equally to this work.
* dr.shannon@cgshe.ubc.ca

**Data Availability Statement:** Due to the highly criminalized and stigmatized nature of this population, anonymized data may be made

## Abstract

### Background

Following a global wave of end-demand criminalization of sex work, the Protection of Communities and Exploited Persons Act (PCEPA) was implemented in Canada, which has implications for the health and safety of sex workers. This study aimed to evaluate the impact of the PCEPA on sex workers' access to health, violence, and sex worker-led services.

### Methods

Longitudinal data were drawn from a community-based cohort of ~900 cis and trans women sex workers in Vancouver, Canada. Multivariable logistic regression examined the independent effect of the post-PCEPA period (2015–2017) versus the pre-PCEPA period (2010–2013) on time-updated measures of sex workers' access to health, violence supports, and sex worker/community-led services.

### Results

The PCEPA was independently correlated with reduced odds of having access to health services when needed (AOR 0.59; 95%CI: 0.45–0.78) and community-led services (AOR 0.77; 95%CI: 0.62–0.95). Among sex workers who experienced physical violence/sexual violence or trauma, there was no significant difference in access to counseling supports post-PCEPA (AOR 1.24; 95%CI: 0.93–1.64).

### Conclusion

Sex workers experienced significantly reduced access to critical health and sex worker/community-led services following implementation of the new laws. Findings suggest end-

available on request subject to the UBC/
Providence Health Ethical Review Board, and
consistent with our funding body guidelines (NIH
and CIHR). Requests should be directed to
info@cgshe.ubc.ca.

**Funding:** This research is supported by the US
National Institutes of Health (NIH)
(R01DA028648), a Canadian Institutes of Health
Research (CIHR) Foundation Grant, and MacAIDS.
SG is partially supported by NIH and a CIHR New
Investigator Award. KS is partially supported by a
Canada Research Chair in Global Sexual Health and
HIV/AIDS, NIH, and Michael Smith Foundation for
Health Research. EA is supported by a CIHR
Doctoral Award. SAS is supported by a NIDA
MERIT Award (R37DA019829). The study funders
had no role in the study design, data collection,
analysis, interpretation, writing of the report, or
decision to submit the paper for publication.

**Competing interests:** The authors have declared
that no competing interests exist.

demand laws may exacerbate and reproduce harms of previous criminalized approaches to sex work in Canada. This study is one of the first globally to evaluate the impact of end-demand approaches to sex work. There is a critical evidence-based need to move away from criminalization of sex work worldwide to ensure full labor and human rights for sex workers. Findings warn against adopting end-demand approaches in other cities or jurisdictions.

## Introduction

Global research and evidence demonstrate that criminal policies and punitive enforcement-based approaches to sex work continue to undermine the health and human rights of sex workers [1–3]. The legal environment has immense potential to shape the wellbeing of those most marginalized in society. Substantial evidence demonstrates that the criminalization of sex work perpetuates widespread forms of violence, stigma, and discrimination that prevent sex workers from seeking or accessing critical health and support services [4–6]. In settings where sex work is criminalized, sex workers are at significantly elevated risk of HIV and other sexually transmitted infections (STIs) driven by social marginalization and increased exposure to workplace violence and abuse [6,7]. The criminalized nature of sex work and related policing practices displace sex workers to more isolated and risker locations and reduce the ability of sex workers to work together or more formally organize due to fear of arrest and police harassment [4,6,8]. Where sex work is criminalized, the ability of sex workers to formally organize or work together is restricted. The hindering of collectivization among sex workers through criminalization is of critical concern given the central importance of community empowerment and enabling sex workers to negotiate safety in the workplace, as well as advocate for human rights, including access to health and safety.

Numerous human rights and public health experts and international bodies, such as the World Health Organization, UNAIDS, and Amnesty International, along with sex work communities worldwide, have strongly endorsed full decriminalization of sex work based on well-established evidence of the harmful impacts of criminalization and enforcement-based approaches [3,9,10]. In 2013, the Supreme Court of Canada struck down three core anti-prostitution laws on the basis that they were a violation of sex workers' constitutional rights [11]; however, Canada's federal government implemented new legislation in 2014, known as the "Protection of Communities and Exploited Persons Act" (PCEPA), which has serious implications for the health and safety of sex workers. Modeled after laws in Norway, Sweden and a number of other European countries, the PCEPA is an end-demand approach that criminalizes new aspects of sex work including communicating for the purpose of selling sex and the purchasing and advertising of sexual services, targeting clients and third parties while leaving the sale of sex legal [12].

Research and legal experts and community have expressed serious concerns regarding end-demand legislation, as it reproduces the same risks and harms of previous criminalization models whereby targeting clients still leads to rushed transactions and improper screening, increasing risk of violence and HIV/STIs [7,13]. The PCEPA also targets third party self-advertising, which has the potential to detrimentally impact sex workers' health and safety, and conflates sex workers with victims of violence and trafficking. Further, the PCEPA focuses on cisgender women sex workers and makes no mention of sex workers who do not identify as cis women (i.e., LGBTQ, men) [12], failing to acknowledge gender and sexual diversity of individuals who sell sex and the unique vulnerabilities faced by gender and sexual minorities [14–16].

One of the explicit goals of end-demand approaches is to increase access to services and supports for sex workers, yet scientific and legal evidence suggest that criminalization may impede access to services [1,2,9,10]. There remains a paucity of empirical research and evidence on the impacts of end-demand approaches globally. Therefore, this study aimed to longitudinally evaluate the impact of the PCEPA on sex workers' access to health, violence, and sex worker/community-led services and supports in Vancouver, Canada.

## Methods

Longitudinal data (2010–2017) were drawn from a community-based, prospective open cohort of over 900 women sex workers in Metro Vancouver known as AESHA (An Evaluation of Sex Workers Health Access). Participants were recruited using time-location sampling [17], with day and late-night outreach to outdoor sex work locations (i.e., streets, alleyways), indoor sex work venues (i.e. massage parlors, micro-brothels, in-call locations), and online. Participatory mapping strategies were conducted to identify work venues, and a weekly mobile van has reached over 100 sex work venues across the city. AESHA includes a diverse experiential team of both current and former sex workers represented across interviewer, outreach, nursing, and coordinator staff since its inception in 2010. AESHA also has a Community Advisory Board of over 15 women's health, sex work and HIV agencies, as well as representatives from health authorities and policy experts.

Eligibility criteria for participants include cis or trans women, 14 years of age or older, who exchanged sex for money within the last 30 days. After providing written informed consent, participants completed interviewer-administered questionnaires and voluntary HIV/STI/HCV serology testing at enrollment and biannually. The questionnaires and clinical components were completed at one of two study offices or at a safe location identified by participants. The main interview questionnaire elicits responses related to socio-demographics (e.g., sexual identity, ethnicity, housing), the work environment (e.g., access to services, safety, policing, incarceration), client characteristics (e.g., types/fees of services, condom use), intimate partners (e.g., cohabitation, financial support), experiences of violence (e.g., childhood abuse, exposure to intimate partner and workplace violence), and drug use patterns. The clinical questionnaire relates to overall physical, mental, and emotional health, and HIV testing and treatment experiences to support education, referral, and linkages with care. The study holds ethical approval through Providence Health Care/University of British Columbia Research Ethics Board. As in previous studies, we have held ethical approval since 2004 to include self-supporting youth aged 14–18 years who are not living with a parent or guardian under the emancipated minor clause, given the critical importance of understanding the needs of vulnerable youth. All participants received an honorarium of $40 CAD at each bi-annual visit for their time, expertise and travel.

### Measures

The main outcomes of interest were time-updated variables for having access to health care when needed and sex worker/community-led services and supports in the last six months. Having access to health services when needed was defined as >75% of the time (responding 'Usually (over 75% of the time)' or 'Always (100% of the time)' to the question 'How often can you get health care services when you need it?'). Utilization of sex worker/community-led services was defined as responding 'yes' to using any sex worker/community-led health or support services, including outreach programs. Access to counseling support for violence/trauma was also examined as an outcome variable among participants who had ever experienced any physical and/or sexual violence and/or lifetime trauma, defined as responding 'yes' to

experiencing any barriers to counseling or therapy for sexual abuse or other trauma or violence.

The primary exposure variable was the post-PCEPA time period (April 2015-August 2017 vs. 2010–2013). Given that the PCEPA was introduced in January 2014 and not officially passed until the end of the year, the year 2014 was dropped from the analyses in order to reduce any potential effects on the outcomes of interest due to variation in the ways in which the laws may have been enforced during this phase. The first three months of 2015 were also excluded to account for outcome measures referring to the preceding six months. Approximately half (53%, n = 452) of participants were interviewed in the pre-PCEPA time period (2010–2013) only, and 14% (n = 117) were interviewed in the post-PCEPA time period (2015–2017) only. One-third (33%, n = 285) of participants were interviewed in both pre- and post-PCEPA time periods. Various other socio-demographic and structural-environmental variables were considered as potential confounders based on the literature and available data collected for the AESHA cohort. Time-fixed variables included gender and/or sexual minority (LGBTQ) and Indigenous ancestry (inclusive of First Nations, Metis, and Inuit). Participant age was updated based on age at baseline and interview date. Primary place of soliciting clients (e.g., street/public spaces, indoor venues/in-call, independent off-street/online), workplace physical and/or sexual violence by clients, police harassment without arrest, any injection and non-injection drug use, and being on any opioid substitution therapy (OST) were considered time-varying and were updated to reflect their occurrence within the last six months.

## Statistical analyses

Descriptive statistics at baseline were calculated for the primary independent variable, the post-PCEPA period, and all potential confounders, stratified by the outcomes of interest. Categorical variables were assessed using Pearson's chi-square test (or Fisher's exact test for small cell counts), and the Wilcoxon rank sum test was used for continuous variables. The relationships between the post-PCEPA period and access to health care and sex worker/community-led supports were examined using bivariate and multivariable logistic regression with generalized estimating equations (GEE) and an exchangeable correlation matrix. Separate multivariable confounder models were fitted to assess the independent relationship between the post-PCEPA period and the outcomes of interest. All analyses were restricted to observations where participants reported engaging in sex work in the last six months; the model for accessing violence supports was further restricted to those who had ever experienced physical and/or sexual violence. A sub-analysis was conducted to examine whether physical and/or sexual workplace violence was affected by the PCEPA; however, these results were not found to be significant. Full models included all hypothesized confounders and were subjected to a manual stepwise approach, whereby variables that altered the association of interest by <5% were systematically removed [18]. Remaining variables were retained as confounders in the final multivariable models. A complete case analysis was used such that observations with any missing data were removed. Two-sided p-values and unadjusted and adjusted odds ratios (ORs and AORs) with 95% confidence intervals (95%CI) for the associations between the post-PCEPA period and the outcomes of interest were generated. All statistical analyses were performed using SAS software version 9.4 (SAS Institute, Cary, NC, USA).

## Results

Of a total 854 participants who completed the baseline questionnaire, 14% (n = 118) reported not having access to health services when needed at baseline and 29% (n = 247) reported not having access at some point during the study. At baseline, 59% (n = 501) reported using a sex

**Table 1. Baseline socio-structural characteristics of sex workers who had access to health services when needed in the last 6 months, compared to those who did not (N = 852).**

| Characteristic | Had access to health services when needed N = 734 (86%) | Did not have access to health services when needed N = 118 (14%) | p-value |
|---|---|---|---|
| Post-PCEPA | 96 (13.1) | 21 (17.8) | 0.167 |
| Age (median, IQR) | 35 (28–42) | 35 (28–43) | 0.747 |
| Gender/sexual minority | 270 (36.8) | 39 (33.1) | 0.428 |
| Indigenous | 288 (39.2) | 43 (36.4) | 0.556 |
| Used non-injection drugs† | 501 (68.3) | 66 (55.9) | 0.007 |
| Used injection drugs† | 310 (42.2) | 36 (30.5) | 0.016 |
| Workplace violence† | 285 (38.8) | 48 (40.7) | 0.752 |
| *On opioid substitution therapy* | | | |
|     No | 257 (35.0) | 41 (34.8) | |
|     Yes | 204 (27.8) | 14 (11.9) | |
|     N/A (never used opioids) | 268 (36.5) | 59 (50.0) | <0.001 |
| *Primary place to solicit clients†* | | | |
|     Street/public space | 384 (52.3) | 48 (40.7) | |
|     Indoor/in-call venue | 194 (26.4) | 51 (43.2) | |
|     Independent/self-advertising | 148 (20.2) | 19 (16.1) | 0.001 |

† In the last 6 months.

worker/community-led health service (70%, n = 596 used these services at some point during the study period). Of a total 683 participants who reported ever experiencing physical and/or sexual violence and/or trauma, 11% (n = 77) reported experiencing barriers to accessing counseling support for violence/trauma at baseline and 31% (n = 209) experienced barriers at some point during the study period. Baseline characteristics among women who had access to health care, sex worker/community-led services and supports, and violence supports are displayed in Tables 1, 2 and 3.

The median age at baseline was 35 years (interquartile range [IQR] = 28–42). At baseline, 36% (n = 310) identified as a gender or sexual minority and 39% (n = 332) as Indigenous, highlighting the overrepresentation of gender and sexual minorities and Indigenous women among sex workers in Vancouver. Among the restricted sample of participants who had ever experienced violence or trauma, 44% (n = 299/683) identified as a gender or sexual minority and 47% (n = 320/683) as Indigenous, and a significantly higher proportion of Indigenous women reported experiencing barriers to counseling (p = 0.016).

Unadjusted and adjusted odds ratios for the associations between the post-PCEPA time period and access to health care, sex worker/community-led services and supports, and counseling for violence/trauma are displayed in Table 4. In final separate multivariable confounder models, the post-PCEPA period was independently associated with significantly reduced odds of having access to health services when needed (AOR 0.59; 95%CI: 0.45–0.78) and sex worker/community-led services and supports (AOR 0.77; 95%CI: 0.62–0.95). Among sex workers who experienced violence or trauma, there was no significant difference in access to counseling supports following implementation of the new laws (AOR 1.24; 95%CI: 0.93–1.64; p = 0.140).

## Discussion

Despite one of the explicit goals of end-demand criminalization approaches being to increase access to services and supports for sex workers, this study found no statistically significant

**Table 2. Baseline socio-structural characteristics of sex workers who utilized sex worker/community-led health and support services in the last 6 months, compared to those who did not (N = 854).**

| Characteristic | Used community services N = 501 (59%) | Did not use community services N = 353 (41%) | p-value |
|---|---|---|---|
| Post-PCEPA | 60 (12.0) | 57 (16.2) | 0.081 |
| Age (median, IQR) | 35 (28–42) | 35 (28–42) | 0.658 |
| Gender/sexual minority | 229 (45.7) | 81 (23.0) | <0.001 |
| Indigenous ancestry | 263 (52.5) | 69 (19.6) | <0.001 |
| Used non-injection drugs† | 443 (88.4) | 125 (35.4) | <0.001 |
| Used injection drugs† | 291 (58.1) | 55 (15.6) | <0.001 |
| Workplace violence† | 257 (51.3) | 79 (22.4) | <0.001 |
| *On opioid substitution therapy* | | | |
| No | 231 (46.1) | 67 (19.0) | |
| Yes | 177 (35.3) | 42 (11.9) | |
| N/A (never used opioids) | 86 (17.2) | 242 (68.6) | <0.001 |
| *Primary place to solicit clients*† | | | |
| Street/public space | 357 (71.3) | 76 (21.5) | |
| Indoor/in-call venue | 31 (6.2) | 215 (60.9) | |
| Independent/self-advertising | 109 (21.8) | 58 (16.4) | <0.001 |

† In the last 6 months.

increase in access to health or sex worker/community-led support services following implementation of the PCEPA in Vancouver, Canada. Rather, findings suggest that after implementation of the new laws, sex workers had reduced access to health and sex worker/community-led supports. To our knowledge, this study is the first to longitudinally evaluate the impact of end-demand legislation on access to health services and supports for sex workers in Canada.

**Table 3. Baseline socio-structural characteristics of sex workers who experienced barriers to receiving counseling for trauma in the last 6 months, compared to those who did not (N = 683)\*.**

| Characteristic | Experienced barriers to support N = 77 (11%) | Did not experience barriers to support N = 606 (89%) | p-value |
|---|---|---|---|
| Post-PCEPA | 12 (15.6) | 84 (13.9) | 0.682 |
| Age (median, IQR) | 32 (28–40) | 35 (28–42) | 0.159 |
| Gender/sexual minority | 33 (42.9) | 266 (43.9) | 0.863 |
| Indigenous ancestry | 46 (59.7) | 274 (45.2) | 0.016 |
| Used non-injection drugs† | 68 (88.3) | 486 (80.2) | 0.054 |
| Used injection drugs† | 41 (53.3) | 293 (48.4) | 0.418 |
| Workplace violence† | 40 (52.0) | 280 (46.2) | 0.302 |
| *On opioid substitution therapy* | | | |
| No | 35 (45.5) | 258 (42.6) | |
| Yes | 23 (29.9) | 188 (31.0) | |
| N/A (never used opioids) | 17 (22.1) | 153 (25.3) | 0.808 |
| *Primary place to solicit clients*† | | | |
| Street/public space | 50 (64.9) | 372 (61.4) | |
| Indoor/in-call venue | 6 (7.8) | 96 (15.8) | |
| Independent/self-advertising | 20 (26.0) | 132 (21.8) | 0.161 |

\*Restricted to workers who reported sexual and/or physical violence or trauma in lifetime.

† In the last 6 months.

**Table 4. Unadjusted and adjusted odds ratios for the effect of the post-PCEPA period (2015–2017 vs. 2010–2013) on sex workers' access to health and sex worker/community-led services and supports in the last 6 months.**

| Health Access Outcomes | Unadjusted Odds Ratio (95% CI) | p-value | Adjusted Odds Ratio (95% CI) | p-value |
|---|---|---|---|---|
| Accessed health services when needed | 0.60 (0.47–0.76) | <0.001 | 0.59 (0.45–0.78)* | <0.001 |
| Utilized community-driven sex work health and support services | 0.73 (0.63–0.85) | <0.001 | 0.77 (0.62–0.95)** | 0.014 |
| Experienced barriers to accessing counseling for sexual abuse, trauma or other violence† | 1.10 (0.86–1.40) | 0.465 | 1.24 (0.93–1.64)*** | 0.140 |

† Restricted to workers who reported sexual and/or physical violence or trauma in lifetime.

* Adjusted for workplace violence, non-injection drug use, and opioid substitution therapy.

** Adjusted for age, Indigeneity, place of solicitation, workplace violence, injection and non-injection drug use, and opioid substitution therapy.

*** Adjusted for age, Indigeneity, place of solicitation, workplace violence and non-injection drug use.

Findings from this study support global calls for full decriminalization of sex work as a critical and necessary structural intervention to improve health and human rights for sex workers and reduce transmission of HIV and other STIs [2,3,7,10]. Existing data suggests that end-demand criminalization that targets clients and third parties, but not sex workers, has been shown to reproduce the risks and harms associated with previous laws criminalizing sex work. For example, a recent study from France found that end-demand laws had detrimental effects on sex workers' safety, health and overall living conditions–worse than the previous laws against soliciting [13]. Qualitative research in Vancouver elucidated the ways in which policing practices that target clients recreate vulnerability to violence by hindering the ability of workers to properly screen clients [7]. Further, the evidence is unequivocal that sex workers who experience physical or sexual violence are less able to negotiate the terms of their transactions and are more likely to experience client condom refusal, significantly increasing risk of HIV/STI transmission [8,19–23]. Marginalized sex workers who experience violence face considerable barriers to accessing counselling for trauma support. The present analysis demonstrates that there was no change in experiencing barriers to accessing counselling for violence or trauma post-PCEPA. This lack of change suggests that end-demand criminalization has failed to address such barriers and may potentially exacerbate the physical and psychological burden among sex workers, especially given that one of the explicit goals of end-demand legislation is to increase access to services and supports for sex workers. Future qualitative work would help to shed more light on sex workers' experiences of barriers to accessing these services pre- vs. post-PCEPA.

Interventions aimed at promoting community empowerment and social cohesion among sex workers can have powerful influences on women's health and safety, as evidenced in lower and middle-income countries [24–26]. However, criminalization, stigma, and a lack of funding to support sex worker-led programs continue to impede collectivization among sex workers [1,24]. Akin to the US PEPFAR anti-prostitution pledge, the PCEPA reduces access to community-led services and jeopardizes funding for and development of critical sex worker-led supports, in addition to further conflating sex work with trafficking [6,10,27,28]. Legislative reform to sex work laws in New Zealand and parts of Australia exemplify the benefits of decriminalizing all aspects of sex work for enabling safer occupational conditions for sex workers, with demonstrated impacts on increased access to health services and improved workplace safety [29–31]. Structural and legal interventions should therefore be guided by the large and growing body of evidence demonstrating that punitive approaches to sex work, including end-demand criminalization such as the PCEPA recently implement in Canada, do not improve health, safety, or access to services for sex workers.

## Strengths and limitations

A major strength of this study is prospective design and use of GEE analyses, which increased statistical power. Sex workers' access to health and support services is likely influenced by a complex set of socio-structural variables, and not all potential confounders could be controlled for in this study. Among the sample restricted to women who experienced violence/trauma, experiencing barriers to counseling may have been underestimated due to the fact that baseline questionnaires prior to September 2014 only asked about sexual violence and not physical violence or trauma. Data were self-reported, which introduces the potential for social desirability and reporting biases, and events that occurred in the past may be subject to recall bias. Given that interviews were conducted in safe and comfortable spaces, alongside the community-based nature of the study represented by experiential interviewers (including current and former sex workers), the likelihood of some biases may have been reduced. Findings may not be generalizable to other sex work populations and settings; however, the study included a wide representation of sex workers from both street and off-street work environments. Women who work more independently (e.g., escorts, online) may have been underrepresented. Community mapping and time-location sampling likely helped to minimize selection bias and ensure a more representative sample of sex workers.

## Conclusions

Findings demonstrate no increase in access to health, violence, and sex worker-led support services post-PCEPA, and rather a reduction in odds of accessing sex worker/community-led supports and health services when needed. End-demand approaches to criminalize sex work may not only reproduce the harms of previous criminalized approaches to sex work in Canada, but may further exacerbate barriers to accessing health and community-led services that have been proven to be key contributors of better health outcomes. There is a critical evidence-based need to move away from criminalized approaches to sex work to ensure full labor and human rights for sex workers, including access to health, social, and legal support services. Findings warn against adopting end-demand approaches in other cities or jurisdictions.

## Acknowledgments

We thank all those who contributed their time and expertise to this project, particularly participants, AESHA community advisory board members and partner agencies, and the AESHA team, including: Jennifer Morris, Jane Li, Minshu Mo, Sherry Wu, Emily Leake, Anita Dhanoa, Meaghan Thumath, Alka Murphy, Jenn McDermid, Tave Cole, Jaime Adams, Roisin Heather, Anna Mathen, Bridget Simpson, Nadina Morin, Desire Tibashoboka, Carly Glanzberg and Maya Henriquez. We also thank Abby Rolston, Peter Vann, Erin Seatter, Jill Chettiar, and Megan Bobetsis for their research and administrative support.

## Author Contributions

**Conceptualization:** Elena Argento, Shira Goldenberg, Steffanie A. Strathdee, Kate Shannon.

**Formal analysis:** Elena Argento, Shira Goldenberg, Melissa Braschel.

**Funding acquisition:** Kate Shannon.

**Investigation:** Elena Argento, Melissa Braschel, Sylvia Machat, Kate Shannon.

**Methodology:** Elena Argento, Melissa Braschel, Steffanie A. Strathdee, Kate Shannon.

**Software:** Melissa Braschel.

**Supervision:** Shira Goldenberg, Steffanie A. Strathdee, Kate Shannon.

**Writing – original draft:** Elena Argento.

**Writing – review & editing:** Elena Argento, Shira Goldenberg, Melissa Braschel, Sylvia Machat, Steffanie A. Strathdee, Kate Shannon.

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
