## [Decision Letter · Decision Letter 0]

14 Aug 2019

PONE-D-19-15535

The impact of end-demand legislation on sex workers’ access to health and sex worker support services: A community-based prospective cohort study in Canada

PLOS ONE

Dear Dr Shannon,

Thank you for submitting your manuscript to PLOS ONE. It is an important piece of research and very timely given the discussions surrounding end-demand policies and I read it with pleasure. I seeked the opinion of a referee and agree with them that it does not fully meet PLOS ONE’s publication criteria. Therefore, we invite you to submit a revised version of the manuscript that addresses the points raised during the review process.

We would appreciate receiving your revised manuscript. To enhance the reproducibility of your results, we recommend that if applicable you deposit your laboratory protocols in protocols.io, where a protocol can be assigned its own identifier (DOI) such that it can be cited independently in the future. For instructions see: http://journals.plos.org/plosone/s/submission-guidelines#loc-laboratory-protocols

We look forward to receiving your revised manuscript.

Kind regards,

Professor Marina Della Giusta

Academic Editor

PLOS ONE

2. Please provide additional details regarding participant consent. In the ethics statement in the Methods and online submission information, please ensure that you have specified (1) whether consent was informed and (2) what type you obtained (for instance, written or verbal). If your study included minors, state whether you obtained consent from parents or guardians. If the need for consent was waived by the ethics committee, please include this information.

**Comments to the Author**

1. Is the manuscript technically sound, and do the data support the conclusions?

Reviewer #1: Yes

2. Has the statistical analysis been performed appropriately and rigorously? 

Reviewer #1: Yes

3. Have the authors made all data underlying the findings in their manuscript fully available?

Reviewer #1: No

4. Is the manuscript presented in an intelligible fashion and written in standard English?

Reviewer #1: Yes

5. Review Comments to the Author

Reviewer #1: Overview: Study evaluates the impact of end demand criminalization of sex work in the form of Protection of Communities and Exploited Persons Act in Canada (PCEPA) on sex workers’ access to health, violence, and sex work led services. The paper is interesting with a clear exposition and important policy implications. Some issues need to be addressed before the paper can be published.

Comments

Introduction:

- While this section provides references to some literature, a deeper engagement with the literature would benefit the readers. It is not always obvious how the literature feeds into the research question. For examples the authors write “Criminalization hinders collectivization among sex workers that is critical to building capacity and enabling sex workers to negotiate safety in the workplace and advocate for the fundamental right to health and equal access to healthcare and support services.” (page 2, lines 39-42). It is not clear why criminalization hinders collectivization, and whether end demand criminalization has a different impact on collectivization in relation to criminalization of the sex worker. Similarly the authors do not explain how PCEPA fails to acknowledge gender diversity of individuals who sell sex (page 3, line 57). It is also not clear how the authors expect differences in effect of PCEPA between say cis-and trans-sex workers or between fsw and msw.

Methods:

- Could the authors report here (rather than later) how many sex workers were interviewed before AND after the passing of the law.

- Could there be any reporting biases because of the recruitment method? While I appreciate that getting a representative sample of sex workers is close to impossible, the biases arising from sampling should be discussed here.

- Was information collected on existence and types of intermediaries (pimps/madams?) to services provided by sex workers? Surely access to health, violence etc is affected by the existence of intermediaries.

Measures:

- The variables ‘access to health services’, ‘utilization of sex worker/community-led services’ and ‘access to counselling support’ are all self-reported variables. Is there any way of checking actual access from service providers?

- Could the authors also control for economic circumstances of the sex worker because presumably the income/wealth may have an effect of access to healthcare? If no questions were asked regarding income, ‘housing’ could be a proxy for socio-economic status. Alternatively, the authors could use ‘fees of services’ as a proxy to earnings of the sex worker (acknowledging that earnings also depend on the presence of intermediaries).

- The time varying covariates like “workplace physical and/or sexual violence” may be affected by the primary exposure variable (pre or post PCEPA). In the result section the authors should report the results without these variables and then add them as robustness checks.

Statistical Analysis:

- Could the authors mention in an appendix what confounders were removed after the manual stepwise approach.

Results:

- Table 1 and 2: Not clear what the first row (Post-PCEPA) indicates- does it mean that 96 women interviewed post-PCEPA had access to health services and 21 didn’t (table 1)? What about the rest of the women interviewed post-PCEPA?

- How do the authors interpret the size of the IORs? Does the magnitude change with added controls?

- Similarly, can the authors shed any light on why the results are not significant for ‘experienced barriers to accessing counselling for sexual abuse, trauma or other violence’? Is this an indication that abused/traumatised sex workers are no more stigmatised in the new legislative regime in comparison to the earlier regime? Why is this result different from the other two outcome variables?

- For the 285 women who were interviewed both pre and post PCEPA time periods the authors could investigate the causal effect of the legislation by conducting a Difference-in-Difference analysis.

Discussion:

- An unanswered question is whether the passing of the end demand criminalization act (i.e. criminalization of the client) affects sex workers’ access to health and sex workers services differently from full criminalization (that of the sex worker). Presumably sex work wasn’t decriminalised before the passing of the law? Is the implication that sex work was tolerated before the passing of the law but has become more stigmatized after the passing of PCEPA?

-

6. PLOS authors have the option to publish the peer review history of their article (what does this mean?). If published, this will include your full peer review and any attached files.

Reviewer #1: No

---

## [Author Response · Author response to Decision Letter 0]

6 Nov 2019

RESPONSE TO REVIEWERS

http://www.journals.plos.org/plosone/s/file?id=wjVg/PLOSOne_formatting_sample_main_body.pdf and

http://www.journals.plos.org/plosone/s/file?id=ba62/PLOSOne_formatting_sample_title_authors_affiliations.pdf

Thank you for your careful review of our manuscript. We have reviewed the PLOS ONE templates and revised to meet the style requirements.

2. Please provide additional details regarding participant consent. In the ethics statement in the Methods and online

submission information, please ensure that you have specified (1) whether consent was informed and (2) what type you obtained (for instance, written or verbal). If your study included minors, state whether you obtained consent from parents or guardians. If the need for consent was waived by the ethics committee, please include this information.

We appreciate this comment and have provided additional text under the methods detailing that all participants provided written informed consent and that our study has held ethical approval since 2004 to include self-supporting youth aged 14–18 years who are not living with a parent or guardian under the emancipated minor clause. Please see additional text under methods as follows:

“As in previous studies, we have held ethical approval since 2004 to include self-supporting youth aged 14–18 years who are not living with a parent or guardian under the emancipated minor clause, given the critical importance of understanding the needs of vulnerable youth.”

Our survey was developed with and by community and includes a number of measures that are validated scales. All variables included in this analysis are discussed in detail under the “measures” section. Please see text on pages 2-4, including details of the following outcome measures and potential confounding variables:

“The main outcomes of interest were time-updated variables for having access to health care when needed and sex worker/community-led services and supports in the last six months. Having access to health services when needed was defined as >75% of the time (responding ‘Usually (over 75% of the time)’ or ‘Always (100% of the time)’ to the question ‘How often can you get health care services when you need it?’). Utilization of sex worker/community-led services was defined as responding ‘yes’ to using any sex worker/community-led health or support services, including outreach programs. Access to counseling support for violence/trauma was also examined as an outcome variable among participants who had ever experienced any physical and/or sexual violence and/or lifetime trauma, defined as responding ‘yes’ to experiencing any barriers to counseling or therapy for sexual abuse or other trauma or violence.

The primary exposure variable was the post-PCEPA time period (April 2015-August 2017 vs. 2010-2013). Given that the PCEPA was introduced in January 2014 and not officially passed until the end of the year, the year 2014 was dropped from the analyses in order to reduce any potential effects on the outcomes of interest due to variation in the ways in which the laws may have been enforced during this phase. The first three months of 2015 were also excluded to account for outcome measures referring to the preceding six months… Various other socio-demographic and structural-environmental variables were considered as potential confounders based on the literature and available data collected for the AESHA cohort. Time-fixed variables included gender and/or sexual minority (LGBTQ) and Indigenous ancestry (inclusive of First Nations, Metis, and Inuit). Participant age was updated based on age at baseline and interview date. Primary place of soliciting clients (e.g., street/public spaces, indoor venues/in-call, independent off-street/online), workplace physical and/or sexual violence by clients, police harassment without arrest, any injection and non-injection drug use, and being on any opioid substitution therapy (OST) were considered time-varying and were updated to reflect their occurrence within the last six months.”

4. Have the authors made all data underlying the findings in their manuscript fully available?

Reviewer #1: No

Please see our data availability statement added to the end of the manuscript, as follows:

“In accordance with the PLOS ONE data access policy, our ethical obligation to research that is of the highest ethical and confidentiality standards, and given the highly criminalized and stigmatized nature of this population, anonymized data may be made available on request subject to the UBC/Providence Health Ethical Review Board, consistent with our funding body guidelines (NIH and CIHR).” 

Reviewer #1: 

Overview: Study evaluates the impact of end demand criminalization of sex work in the form of Protection of

Communities and Exploited Persons Act in Canada (PCEPA) on sex workers’ access to health, violence, and sex work led services. The paper is interesting with a clear exposition and important policy implications. Some issues need to be addressed before the paper can be published.

Thank you for your comprehensive review of our manuscript and we believe that the revisions as outlined below have strengthened the paper considerably.

Comments

Introduction:

- While this section provides references to some literature, a deeper engagement with the literature would benefit the readers. It is not always obvious how the literature feeds into the research question. For examples the authors write “Criminalization hinders collectivization among sex workers that is critical to building capacity and enabling sex workers to negotiate safety in the workplace and advocate for the fundamental right to health and equal access to healthcare and support services.” (page 2, lines 39-42). It is not clear why criminalization hinders collectivization, and whether end demand criminalization has a different impact on collectivization in relation to criminalization of the sex worker. Similarly the authors do not explain how PCEPA fails to acknowledge gender diversity of individuals who sell sex (page 3, line 57). It is also not clear how the authors expect differences in effect of PCEPA between say cis-and trans-sex workers or between fsw and msw.

We appreciate this helpful comment. We have included additional text in the introduction clarifying how criminalization and policing practices hinder collectivization among sex workers and the ability of sex workers to protect against violence in the workplace. We have also expanded on how the PCEPA fails to acknowledge gender diversity by focusing solely on women sex workers. While it is beyond the scope of this paper’s introduction to discuss the potential differential impacts of the PCEPA on specific genders, we have also included text and additional references highlighting that gender/sexual minorities face unique vulnerabilities. We have revised as follows:

“The criminalized nature of sex work and related policing practices displace sex workers to more isolated and risker locations and reduce the ability of sex workers to work together or more formally organize due to fear of arrest and police harassment [4,6,8]. Where sex work is criminalized, the ability of sex workers to formally organize or work together is restricted. The hindering of collectivization among sex workers through criminalization is of critical concern given the central importance of community empowerment and enabling sex workers to negotiate safety in the workplace, as well as advocate for human rights, including access to health and safety.”

“Research and legal experts and community have expressed serious concerns regarding end-demand legislation, as it reproduces the same risks and harms of previous criminalization models whereby targeting clients still leads to rushed transactions and improper screening, increasing risk of violence and HIV/STIs [7,13]. The PCEPA also targets third party self-advertising, which has the potential to detrimentally impact sex workers’ health and safety, and conflates sex workers with victims of violence and trafficking. Further, the PCEPA focuses on cisgender women sex workers and makes no mention of sex workers who do not identify as cis women (i.e., LGBTQ, men) [12], failing to acknowledge gender and sexual diversity of individuals who sell sex and the unique vulnerabilities faced by gender and sexual minorities [14–16].”

Methods:

- Could the authors report here (rather than later) how many sex workers were interviewed before AND after the passing of the law.

Thank you for this suggestion. We have moved this information up to the Methods section.

- Could there be any reporting biases because of the recruitment method? While I appreciate that getting a representative sample of sex workers is close to impossible, the biases arising from sampling should be discussed here.

We appreciate this comment. We have included the following text under the section “strengths and limitations” discussing reporting and recall biases, the community-based nature of our study and interview process, as well as community mapping and time-location sampling that was used for recruitment which helped to ensure a more representative sample:

“Data were self-reported, which introduces the potential for social desirability and reporting biases, and events that occurred in the past may be subject to recall bias. Given that interviews were conducted in safe and comfortable spaces, alongside the community-based nature of the study represented by experiential interviewers (including current and former sex workers), the likelihood of some biases may have been reduced. Findings may not be generalizable to other sex work populations and settings; however, the study included a wide representation of sex workers from both street and off-street work environments. Women who work more independently (e.g., escorts, online) may have been underrepresented. Community mapping and time-location sampling likely helped to minimize selection bias and ensure a more representative sample of sex workers.”

- Was information collected on existence and types of intermediaries (pimps/madams?) to services provided by sex workers? Surely access to health, violence etc is affected by the existence of intermediaries.

Thank you for this question and comment. We have recently published a paper on the impact of third parties (managers, venue owners, security, etc.) demonstrating that engagement with these types of intermediaries increased sex workers’ access to occupational health and safety services (e.g., mobile condom distribution and community-led services). Further, this study demonstrated that end-demand criminalization was associated with decreased access to third party services. Please see: Bronwyn McBride, Shira M. Goldenberg, Alka Murphy, Sherry Wu, Melissa Braschel, Andrea Krüsi, and Kate Shannon, 2019: Third parties (venue owners, managers, security, etc.) and access to occupational health and safety among sex workers in a Canadian setting: 2010-2016. AJPH 109, 792-798.

Measures:

- The variables ‘access to health services’, ‘utilization of sex worker/community-led services’ and ‘access to counselling support’ are all self-reported variables. Is there any way of checking actual access from service providers?

Thank you for this comment. Our study drew on available data from the AESHA questionnaire initiated in 2010, allowing us to utilize longitudinal data based on self-report. While future studies could certainly investigate service utilization based on service provider databases, we note that there is a large body of research that suggests that sex workers are often reluctant to disclose engagement in sex work due to stigma and criminalization and thus would be highly underrepresented in external databases. For this study we were interested in whether or not sex workers reported that they had access to health care when needed and if they accessed sex worker/community-led services and outreach programs. Using questionnaire data, we were further able to restrict the sample to women who had ever experienced physical and/or sexual violence or trauma and examine the association between the new laws and whether or not sex workers reported that they had experienced barriers to counselling or therapy for violence/trauma. Importantly, our nursing, outreach, and interviewer teams are represented by experiential (current and former sex workers) individuals and those with considerable rapport with the community. As such, reporting biases relating to self-reported data were likely minimized. The AESHA cohort also has an annual retention rate of >90%. 

- Could the authors also control for economic circumstances of the sex worker because presumably the income/wealth may have an effect of access to healthcare? If no questions were asked regarding income, ‘housing’ could be a proxy for socioeconomic status. Alternatively, the authors could use ‘fees of services’ as a proxy to earnings of the sex worker (acknowledging that earnings also depend on the presence of intermediaries).

We appreciate this helpful comment. We tried running all multivariable confounder models including the variable for average monthly income. This variable was not retained as a confounder in any of the three models and did not change the results.

- The time varying covariates like “workplace physical and/or sexual violence” may be affected by the primary exposure variable (pre or post PCEPA). In the result section the authors should report the results without these variables and then add them as robustness checks.

Thank you for this comment. We ran sub-analysis to determine if workplace violence could be affected by the PCEPA and thus be mediating the effect, and this was not found to be significant. Please see additional text under the Statistical Analyses section as follows:

“A sub-analysis was conducted to examine whether physical and/or sexual workplace violence was affected by the PCEPA; however, these results were not found to be significant.”

Statistical Analysis:

- Could the authors mention in an appendix what confounders were removed after the manual stepwise approach.

Thank you for this question. We have included in Table 4 notes which variables were adjusted for. All of the other confounders are listed in tables 1-3. As mentioned above in response to your other query, we also tried running the models with the variable for average monthly income; however, this variable was not retained in any of the three models. 

Results:

- Table 1 and 2: Not clear what the first row (Post-PCEPA) indicates- does it mean that 96 women interviewed post-PCEPA had access to health services and 21 didn’t (table 1)? What about the rest of the women interviewed post-PCEPA?

Thank you for your comment and request for clarification of Tables 1 and 2. The first row in Table 1 for example, indicates that out of the 734 women who said they had access to health services when needed at baseline, 96/734 (or 13%) had access while 21/118 (or 17.8%) didn’t. As there were 117 women interviewed post-PCEPA, these numbers include all women who were interviewed post-PCEPA (96+21=117). Table 2 is organized in the same way, with 60 women reporting using community services after the new laws were implemented and 57 reporting not using community services post-PCEPA (60+57=117). We have included the total number of women interviewed post-PCEPA (n=117) in the methods, as suggested in your query above. 

- How do the authors interpret the size of the IORs? Does the magnitude change with added controls?

We appreciate this comment. As described in the results and discussion, after controlling for potential confounders in the final separate multivariable confounder models, sex workers had significantly reduced odds of having access to health services when needed (AOR 0.59; 95%CI: 0.45-0.78) and sex worker/community-led services and supports (AOR 0.77; 95%CI: 0.62-0.95); among sex workers who experienced violence or trauma, there was no significant difference in access to counseling supports following implementation of the new laws (AOR 1.24; 95%CI: 0.93-1.64; p=0.140). As such, this study found no statistically significant increase in access to services and supports following implementation of the PCEPA in Vancouver. Please see text included in the methods/statistical analyses section clarifying the manual stepwise approach for each confounder model, whereby variables that altered the association by <5% were systematically removed. Please also see text included in the discussion section where we expand upon why end-demand laws may be reproducing harms associated with previous criminalization models, and why decriminalization of sex work is a necessary structural intervention to improve access to health, as demonstrated by our study and a growing body of research and evidence worldwide. 

- Similarly, can the authors shed any light on why the results are not significant for ‘experienced barriers to accessing

counselling for sexual abuse, trauma or other violence’? Is this an indication that abused/traumatised sex workers are no more stigmatised in the new legislative regime in comparison to the earlier regime? Why is this result different from the other two outcome variables?

Thank you for this comment. While the results were not significant for experiencing barriers to accessing counselling for violence/trauma post-PCEPA, the lack of change in this variable is noteworthy, given that one of the explicit goals of end-demand legislation is to increase access to services and supports for sex workers. As such, this null finding indicates that the new laws did not improve access to support among sex workers who have experienced physical/sexual violence and trauma. Please see text included in the Discussion as follows:

“Despite one of the explicit goals of end-demand criminalization approaches being to increase access to services and supports for sex workers, this study found no statistically significant increase in access to health or sex worker/community-led support services following implementation of the PCEPA in Vancouver, Canada. Rather, findings suggest that after implementation of the new laws, sex workers had reduced access to health and sex worker/community-led supports…”

“Further, the evidence is unequivocal that sex workers who experience physical or sexual violence are less able to negotiate the terms of their transactions and are more likely to experience client condom refusal, significantly increasing risk of HIV/STI transmission [8,19–23]. Marginalized sex workers who experience violence face considerable barriers to accessing counselling for trauma support. The present analysis demonstrates that there was no change in experiencing barriers to accessing counselling for violence or trauma post-PCEPA. This lack of change suggests that end-demand criminalization has failed to address such barriers and may potentially exacerbate the physical and psychological burden among sex workers, especially given that one of the explicit goals of end-demand legislation is to increase access to services and supports for sex workers. Future qualitative work would help to shed more light on sex workers’ experiences of barriers to accessing these services pre – vs. post-PCEPA.” 

- For the 285 women who were interviewed both pre and post PCEPA time periods the authors could investigate the causal effect of the legislation by conducting a Difference-in-Difference analysis.

We appreciate your suggestion to conduct a difference-in-difference analysis. However, this approach requires a reference group who are not exposed to the policy intervention. Though this would be a very informative analysis, all of the participants in our study are exposed to PCEPA, so we have to rely on a pre-post design here. 

Discussion:

- An unanswered question is whether the passing of the end demand criminalization act (i.e. criminalization of the client) affects sex workers’ access to health and sex workers services differently from full criminalization (that of the sex worker). Presumably sex work wasn’t decriminalised before the passing of the law? Is the implication that sex work was tolerated before the passing of the law but has become more stigmatized after the passing of PCEPA?

We appreciate this comment and suggestion to clarify how the impact of end demand criminalization differs from previous models of criminalization of sex work on access to services and supports. We have included text in the introduction clarifying that the PCEPA criminalizes new aspects of sex work (communicating for the purpose of selling sex and purchasing and advertising of sexual services) targeting clients and third parties while leaving the sale of sex legal. In the Discussion section, we have elaborated on our study findings which build on mounting evidence globally to suggest that end-demand approaches continue to undermine the health and safety of sex workers and reproduce the same risks and harms seen in other criminalized approaches to sex work. Sex work in Canada was not decriminalized prior to the PCEPA, but rather other aspects of sex work were criminalized; the new end-demand laws continue to criminalize sex work with an emphasis on targeting clients, which has been shown to have a detrimental impact on sex workers’ access to health services and supports, as seen in the present study, as well as overall safety and wellbeing. As such, our study supports global calls for full decriminalization of sex work. Please see text throughout the Discussion section, including the following:

“Existing data suggests that end-demand criminalization that targets clients and third parties, but not sex workers, has been shown to reproduce the risks and harms associated with previous laws criminalizing sex work. For example, a recent study from France found that end-demand laws had detrimental effects on sex workers’ safety, health and overall living conditions – worse than the previous laws against soliciting [13]. Qualitative research in Vancouver elucidated the ways in which policing practices that target clients recreate vulnerability to violence by hindering the ability of workers to properly screen clients [7]. Further, the evidence is unequivocal that sex workers who experience physical or sexual violence are less able to negotiate the terms of their transactions and are more likely to experience client condom refusal, significantly increasing risk of HIV/STI transmission [8,19–23]. Marginalized sex workers who experience violence face considerable barriers to accessing counselling for trauma support. The present analysis demonstrates that end-demand criminalization has failed to address such barriers and may potentially exacerbate the physical and psychological burden among sex workers.”

“Akin to the US PEPFAR anti-prostitution pledge, the PCEPA reduces access to community-led services and jeopardizes funding for and development of critical sex worker-led supports, in addition to further conflating sex work with trafficking [6,10,27,28]. Legislative reform to sex work laws in New Zealand and parts of Australia exemplify the benefits of decriminalizing all aspects of sex work for enabling safer occupational conditions for sex workers, with demonstrated impacts on increased access to health services and improved workplace safety [29–31]. Structural and legal interventions should therefore be guided by the large and growing body of evidence demonstrating that punitive approaches to sex work, including end-demand criminalization such as the PCEPA recently implement in Canada, do not improve health, safety, or access to services for sex workers.”

---

## [Editor Report · Decision Letter 1]

13 Nov 2019

The impact of end-demand legislation on sex workers’ access to health and sex worker-led services: A community-based prospective cohort study in Canada

PONE-D-19-15535R1

Dear Dr. Shannon,

We are pleased to inform you that your revised manuscript has been judged suitable for publication and will be formally accepted for publication once it complies with all outstanding technical requirements. We think the manuscript has indeed significantly improved after your revisions and will provide a helpful contribution to the literature.

With kind regards,

Professor Marina Della Giusta

Academic Editor

PLOS ONE

---

## [Editor Report · Acceptance letter]

19 Nov 2019

PONE-D-19-15535R1 

The impact of end-demand legislation on sex workers’ access to health and sex worker-led services: A community-based prospective cohort study in Canada 

Dear Dr. Shannon:

I am pleased to inform you that your manuscript has been deemed suitable for publication in PLOS ONE. Congratulations! Your manuscript is now with our production department. 

With kind regards,

on behalf of

Associate Professor Marina Della Giusta 

Academic Editor

PLOS ONE